# UTX is an escape from X-inactivation tumor-suppressor in B cell lymphoma

Xiaoxi Li [1,2], Yanli Zhang[1], Liting Zheng[3], Mingxian Liu[1], Charlie Degui Chen[3] & Hai Jiang [1]

To explain the excess cancer rate in males, several candidates for "escape from X-inactivation tumor-suppressor" (EXITS) were recently identified. In this report we provide direct experimental evidence supporting UTX's role as an EXITS gene. Using a mouse lymphoma model, we show clear dosage effect of UTX copy number during tumorigenesis, which strongly supports the EXITS theory. Importantly, UTX deletion not only accelerates lymphomagenesis, it also strongly promotes tumor progression. UTX-knockout tumors are more aggressive, showing enhanced brain dissemination and formation of blood vessels. Efnb1 is overexpressed in UTX KO tumors and can lead to such phenotypes. In human patients, lymphomas with low UTX expression also express high levels of Efnb1, and cause significantly poor survival. Lastly, we show that UTX deficiency renders lymphoma sensitive to cytarabine treatment. Taken together, these data highlight UTX loss's profound impacts on tumor initiation and drug response.

---

[1] State Key Laboratory of Cell Biology, Key Laboratory of Systems Biology, Innovation Center for Cell Signaling Network, CAS Center for Excellence in Molecular Cell Science, Shanghai Institute of Biochemistry and Cell Biology, Chinese Academy of Sciences, University of Chinese Academy of Sciences, 320 Yueyang Road, 200031 Shanghai, China. [2] Jiangsu Key Laboratory of Medical Science and Laboratory Medicine. School of Medicine, Jiangsu University, ZhenjiangJiangsu, China. [3] Present address: State Key Laboratory of Molecular Biology, Shanghai Key Laboratory of Molecular Andrology, Innovation Center for Cell Signaling Network, CAS Center for Excellence in Molecular Cell Science, Shanghai Institute of Biochemistry and Cell Biology, Chinese Academy of Sciences, University of Chinese Academy of Sciences, 320 Yueyang Road, 200031 Shanghai, China. Correspondence and requests for materials should be addressed to H.J. (email: hai@sibcb.ac.cn)

Ubiquitously transcribed tetratricopeptide repeat X-linked protein (UTX) (also known as KDM6A) is an epigenetic regulator that functions as a demethylase for histone H3K27[1]. Through recent cancer genome sequencing studies, UTX is found to be commonly mutated or deleted in various types of human tumor[2–7]. According to the COSMIC database (the Catalogue of Somatic Mutations in Cancer[8]), nearly 40% of mutations found on UTX are nonsense or frameshift mutations, which abolish UTX expression. This suggests UTX could act as a tumor suppressor.

UTX is an essential gene. Female UTX$^{-/-}$ mice die at E9.5, and only a small fraction of UTX$^{-/Y}$ male mice survive to adulthood, which indicates UTY could compensate for UTX loss during development[9]. The unavailability of UTX$^{-/-}$ mice, as well as the potential compensation by UTY complicates the study of UTX's role as tumor suppressor. Using hematopoietic stem cell (HSC) from surviving UTX$^{-/Y}$ mice, Ntziachristos et al. showed that UTX deficiency in male HSCs accelerates Notch1-induced T cell acute lymphoblastic leukemia (T-ALL), when transplanted into recipient mice[10]. Another study, using similar ex vivo models, showed that shRNA-mediated knockdown of UTX also accelerated Notch1-induced T-ALL[11]. These studies highlighted the tumor suppressor role of UTX during leukemogenesis. However, in these studies, the dosage effect of UTX, the potential compensation by UTY, as well as UTX's impacts on tumor progression remain largely unclear.

Interestingly, although located on X-chromosome, UTX escapes from X-chromosome inactivation, and both copies of UTX are found to express in females[12,13]. Therefore, it is proposed that for females, mutation or deletion of both copies of UTX is needed to functionally inactivate this potential tumor suppressor, whereas in males inactivating one copy of UTX will suffice. Through comprehensive analysis of gene mutation status of human cancers, several genes, including UTX, were recently identified as candidates for "escape from X-inactivation tumor-suppressor" (EXITS), which could explain the excess cancer incidence in males[13,14]. To stringently test this idea, we argue that it is necessary to employ tissue-specific UTX-knockout mice, so that the aforementioned dosage effect could be addressed with UTX$^{+/-}$ and UTX$^{-/-}$ female mice. Also, by analyzing the UTX$^{-/Y}$ mice, we could ask whether UTY could functionally compensate for UTX during tumorigenesis. The answer to the latter question is also important, because if UTY offers significant compensation for UTX during tumorigenesis, then UTX's importance as an X-chromosome coded tumor suppressor would diminish.

In this study, utilizing a mouse lymphoma model and conditional UTX-knockout mice, we addressed these questions. Importantly, we showed that UTX loss not only promotes tumor formation, it also strongly enhances the aggressiveness of lymphoma, as evidenced by brain dissemination and formation of blood vessels, through upregulation of Efnb1. We also observed that UTX deficiency confers enhanced sensitivity to the anticancer drug cytarabine, suggesting possible approaches to targeting UTX-deficient tumors.

## Results

**UTX deficiency leads to poor survival in human lymphoma**. To address the dosage effects of UTX and UTY's potential compensation during tumorigenesis, we utilized UTX$^{f/f}$ and UTX$^{f/y}$ mice. We chose to cross these mice with CD19-CRE mice to generate B-lymphocyte specific UTX knockout based on several observations. First, UTX is recurrently mutated in various forms of B cell lymphoma and leukemia[5,15]. Analysis of the TCGA Copy Number Portal[16] (http://portals.broadinstitute.org/tcga/home)

also indicated that about 10% of diffuse large B cell lymphoma samples exhibit deletion of the UTX gene. Second, cancer gene expression analysis[17] using human B cell lymphoma clinical database (Lenz Staudt Lymphoma GSE10846[18]) suggests that low expression level of UTX is associated with significantly poor survival (Fig. 1a). Male patients are enriched in high-risk group with low UTX expression, while the female patients are enriched in low-risk group with high UTX expression (Fig. 1b). The sex-difference on prognosis and its relationship to UTX expression level suggest that gender could be a key factor for lymphomagenesis and progression, potentially owing to the putative EXITS genes. Therefore, the B cell lineage provides an ideal setting for studying UTX's contribution to sex-difference of cancer incidence observed in humans. It also provides a good setting to study whether and how UTX loss impacts tumor aggressiveness, given that low expression level of UTX is associated with poor patient survival.

**UTX acts as an EXITS during Eµ-Myc-induced lymphomagenesis**. Given the above evidence, we crossed CD19-CRE with UTX$^{f/f}$ mice and generated B-lymphocyte specific UTX knockouts. Deletion of UTX in B cells alone did not induce lymphoma, either in male or female mice till 200 days (Fig. 1c). Of note, the oldest batch of CD19-CRE;UTX$^{-/-}$ mice created during this project remained tumor free till 18 month. To further address UTX's role in lymphomagenesis, we crossed CD19-CRE;UTX$^{f/f}$ mice with Eµ-Myc transgenic mice, a well-established mouse lymphoma model. When crossing Eµ-Myc to CD19-CRE;UTX$^{f/f}$ mice, all genotypes were born at expected mendelian rate (Supplementary Fig. 1). This enabled us to examine lymphomagenesis in male and female mice with different UTX and UTY status.

In male mice, the vast majority of Eµ-Myc;UTX$^{+/y}$ genotype died of lymphoma by 150 days, whereas deletion of UTX in this background further accelerated lymphomagenesis ($p < 0.0001$, log-rank test) (Fig. 1c). In female mice, only a small portion of Eµ-Myc;UTX$^{+/+}$ genotype died of lymphoma by 200 days. Deletion of one copy of UTX in this background accelerated lymphomagenesis, with 50% of mice died of lymphoma by 200 days ($p = 0.0244$, log-rank test) (Fig. 1d). Deletion of both copies of UTX in this background further accelerated lymphomagenesis, with nearly 90% of mice died of lymphoma by 100 days ($p < 0.0001$, log-rank test) (Fig. 1d). These results demonstrated UTX's tumor suppressive role in lymphoma, both in male and female settings.

Importantly, our data confirmed the dosage effect of UTX during tumorigenesis. Comparing to single copy deletion of UTX, deletion of both copies of UTX resulted in much faster rate of lymphomagenesis ($p < 0.0001$, log-rank test) (Fig. 1e). This strongly supported the theory proposed by Van der Meulen et al., and showed that UTX is a key candidate for the EXITS genes.

In addition, we observed that Eµ-Myc;UTX$^{-/y}$ mice developed lymphoma at a relatively slower pace than Eµ-Myc;UTX$^{-/-}$ mice ($p < 0.0001$, log-rank test) (Fig. 1e). This suggests UTY may provide some compensation for UTX loss during tumorigenesis. However, this compensation is only minimal, given that Eµ-Myc;UTX$^{-/y}$ mice still developed lymphoma significantly faster than Eµ-Myc;UTX$^{+/-}$ single knockout mice ($p = 0.0010$, log-rank test) (Fig. 1f). Notably, Eµ-Myc;UTX$^{-/y}$ mice developed lymphoma with complete penetrance, whereas only 50% of Eµ-Myc;UTX$^{+/-}$ single knockout mice developed lymphoma, further arguing that UTY could not provide significant compensation for UTX during tumorigenesis (Fig. 1f).

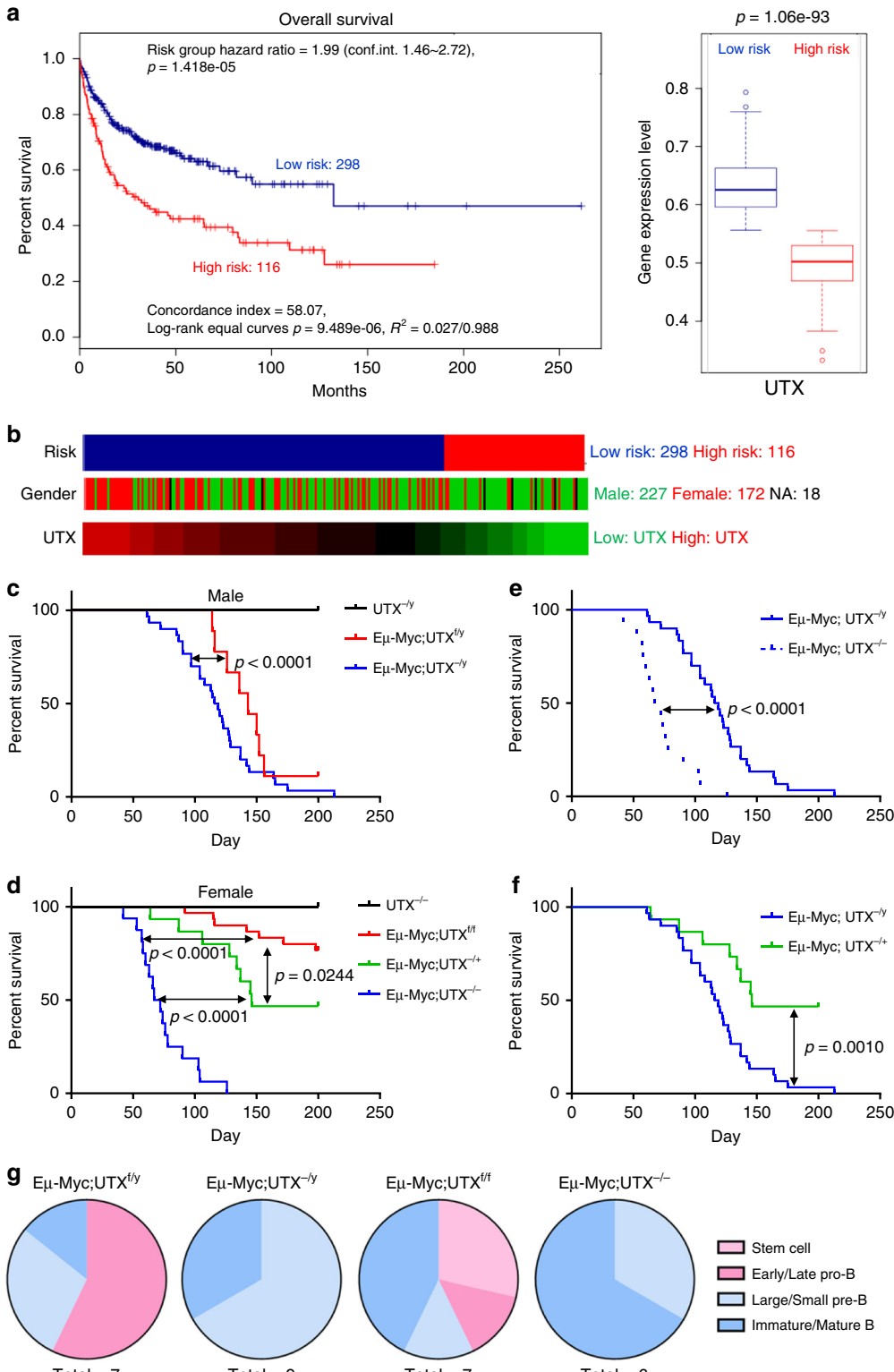

**Fig. 1** UTX acts as an 'escape from X-inactivation tumor-suppressor' during Eμ-Myc-induced lymphomagenesis. **a** Expression level of UTX in human lymphoma were correlated with risk and overall survival time. UTX expression status was used as the variable to test if it can predict patient survival. Green and red lines indicated low-risk ($n = 298$) and high-risk ($n = 116$) groups, respectively. Risk grouping was conducted through an optimization algorithm. Analysis was performed by SurvExpress. High-risk group was correlated with low expression of UTX. **b** Distribution of male and female patients in risk groups separated by UTX expression. Male were enriched in low expression of UTX indicated low risk, poor prognosis. Female were enriched in high expression of UTX indicated low risk, good prognosis. **c** Kaplan–Meier curves indicating the survival of male mice with each cohort, UTX$^{-/y}$ mice ($n = 19$, black), Eμ-Myc;UTX$^{+/y}$ mice ($n = 9$, red), Eμ-Myc;UTX$^{-/y}$ mice ($n = 30$, blue). **d** Kaplan–Meier curves indicating the survival of female mice with each cohort, UTX$^{-/-}$ mice ($n = 19$, black), Eμ-Myc mice ($n = 30$, red), Eμ-Myc;UTX$^{-/+}$ mice ($n = 15$, green), Eμ-Myc;UTX$^{-/-}$ mice ($n = 16$, blue). **e** Comparison of Kaplan–Meier curves of Eμ-Myc;UTX$^{-/-}$ and Eμ-Myc;UTX$^{-/y}$. **f** Comparison of Kaplan–Meier curves of Eμ-Myc;UTX$^{-/y}$ and Eμ-Myc;UTX$^{+/-}$. Statistical significance was determined using log-rank test for panel C to F. **g** Distribution of B cell differentiation stages for lymphomas from each genotype

**UTX deficiency enhances the aggressiveness of lymphoma**. To study the type and developmental stages of lymphoma cells in our mouse models, we used various cell surface markers to perform FACS analysis of lymphoma cells. The results showed that in both male and female UTX-knockout Eμ-Myc mice, lymphomas originated exclusively from later stage B cells (pre-B and immature/mature B cell), whereas UTX wild-type Eμ-Myc lymphomas also originated from early stage B cells (stem cell and pro-B cell) (Fig. 1g, Supplementary Fig. 2).

Analysis of these lymphoma-bearing mice showed classic lymphoma distributions, such as significant bulges of peripheral lymph nodes, including inguinal, brachial, axillary, and cervical lymph nodes, regardless of gender or UTX status (Fig. 2a). These mice also showed enlarged spleens, and UTX-knockout Eμ-Myc

mice showed more pronounced abnormality in germinal center (Fig. 2b–d, Supplementary Fig. 3).

Interestingly, in contrast to their UTX wild-type counterparts, UTX-knockout Eμ-Myc mice exhibited multiple signs of advanced lymphoma, including brain metastasis, which caused skull bulges, and huge thymic/mediastinal tumors in the chest cavity (Fig. 2e–i and Supplementary Fig. 4a, b). Further analysis showed that significant number of lymphoma cells colonized the space between brain and skull (Supplementary Fig. 4a), and surface marker analysis of these cells and tumor cells in chest cavities confirmed that they are of B cell origin. Of note, such mice with brain and thymic/mediastinal dissemination still exhibited typical lymphoma phenotypes. Nearly all lymph nodes in these mice developed solid tumors. This suggests the diseases

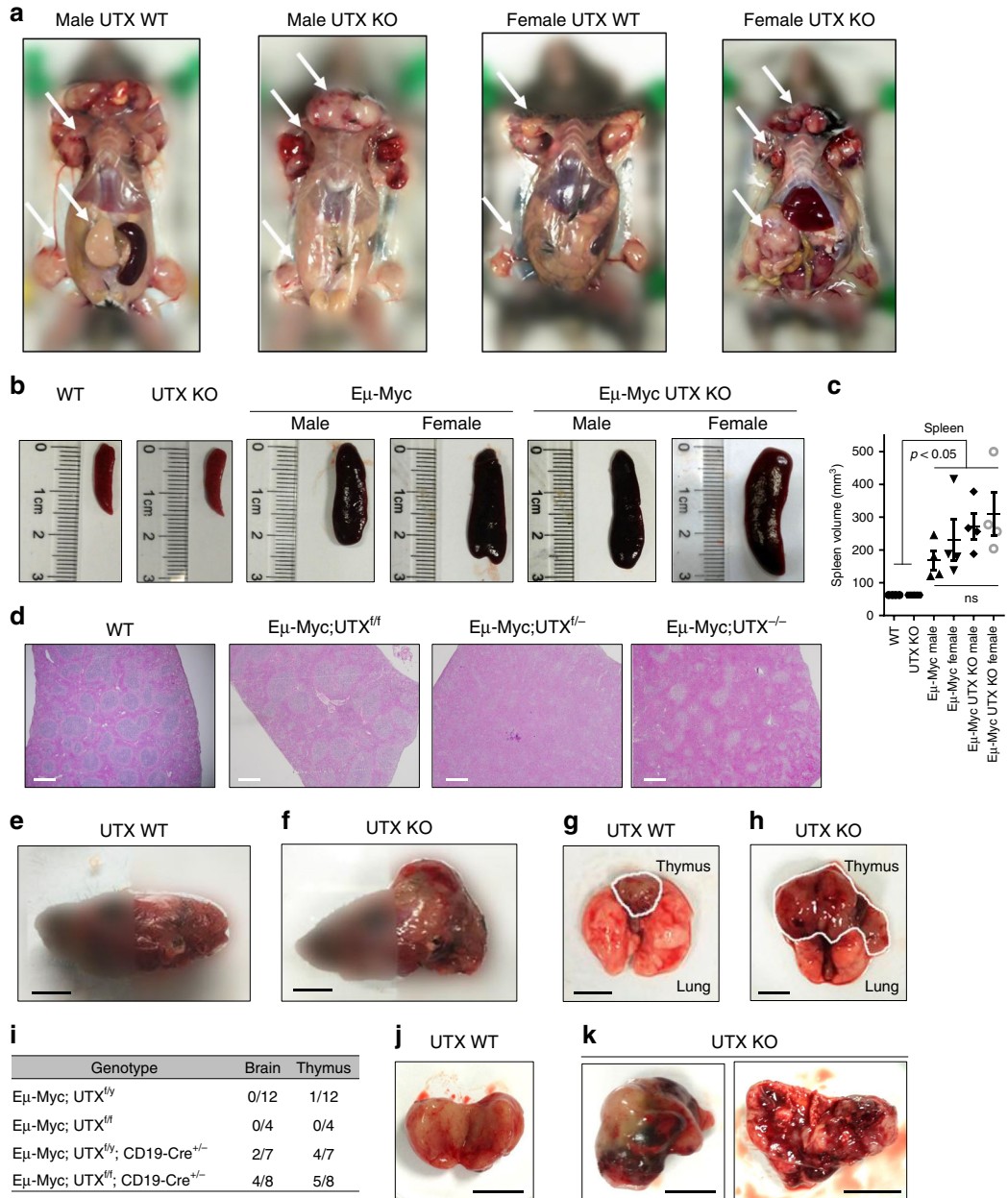

**Fig. 2** Lymphomas in Eμ-MYC;UTX KO mice exhibited enhanced aggressiveness. **a** Representative images of lymphoma-bearing mice. **b** Representative images of spleens for each genotype. **c** Spleen volume of each genotype. Shown are mean ± SEM, $n = 4$ for each genotype. **d** Representative HE staining images of spleens of each genotype, Bar, 500 μm. **e, f** Normal skull in UTX WT mice and skull bulges in UTX KO mice. **g, h** Normal thymus in UTX WT mice and enlarged thymus in UTX KO mice. **i** Summary of extranodal lymphoma in brain and thymus. **j, k** Representative images of lymphomas in UTX WT mice (**j**) and UTX KO mice (**k**). In Panel **k**, image in the right is a cut-open lymphoma showing blood vessels. Bar, 5 mm

in these mice are still lymphomas, but not leukemia, which usually do not develop solid tumors.

Moreover, compared to lymphomas generated in UTX wild-type mice, UTX-knockout lymphomas often showed visible blood infusion (Fig. 2j–k and Supplementary Fig. 4c, d). Taken together, these observations suggest that in addition to lymphoma initiation, UTX loss also significantly promotes the progression of lymphomas, as evidenced by dissemination and blood vessel formation. Interestingly, this is consistent with the observation that in human lymphomas, low UTX expression defines the high-risk group and is associated with poor survival (Fig. 1a, b). Furthermore, in male and female lymphoma patients, although the percentages of UTX-low tumors are different, UTX-low status significantly predicts poor survival in both genders. This indicate that UTX level is independently prognostic in human lymphomas (Supplementary Fig. 5).

**UTX KO impacts B cell development and cell adhesion**. To understand how UTX loss may affect lymphoma initiation and progression, we performed transcriptome analysis of littermate UTX wild-type and knockout lymphomas. Given that in male mice UTY may partially compensate for UTX loss, we analyzed lymphomas from UTX wild-type and UTX-knockout female mice. Pathway analysis suggested that in UTX KO lymphomas, the most significantly impacted pathways included those affecting B cell biology (primary immunodeficiency, hematopoietic cell lineage, antigen processing, and presentation) (Fig. 3a, b), which probably reflects the different developmental stages of such tumors (Fig. 1g). In addition, cell adhesion pathway was also significantly impacted by UTX status (Fig. 3a, b, Supplementary Table 1–3). Specifically, genes affected by UTX knockout included Dntt, Rag1, VpreB1, Igll1, Fcmr, and Blk, which are involved in B cell differentiation and cellular fitness, and Bok, a proapoptotic gene commonly deleted in human tumors, as well as genes with potential roles in metastasis and angiogenesis, such as Efnb1. To further confirm the differential expression of these genes, we used qPCR to quantify their mRNA level in UTX WT and UTX KO lymphoma samples, and the results were consistent with our transcriptome analysis. For these genes, the expression levels often varied by at least 100 fold between UTX wild-type and UTX-knockout lymphomas (Fig. 3c).

Among the upregulated genes in UTX-knockout lymphomas, Blk (B Lymphocyte Kinase) is a proto-oncogene belonging to the Src kinase family. It functions downstream of the pre-B cell receptor, and is involved in B cell activation and expansion. Blk enhances the proliferation of B progenitor cells in vivo, and Blk by itself mimics the signaling of pre-B cell receptor (BCR)[19], whose activity was recently shown to enhance the fitness of Myc-driven lymphoma cells[20]. Moreover, aberrant activity of Blk in B cell lineage induced B lymphoid tumors[21]. Therefore, in UTX-knockout settings, overexpressed Blk may explain the enhanced lymphomagenesis phenotype.

**Expression of Efnb1 enhances tumor aggressiveness in vivo**. Among the highly overexpressed genes in UTX-knockout lymphomas, Efnb1 may contribute to the enhanced dissemination and blood vessel formation phenotype. Efnb1 belongs to the ephrins family. Previous reports showed that ephrins family genes, along with ephrin receptors, play important roles in tumor invasion and angiogenesis[22–25].

To date, Efnb1 has not been studied in a metastasis setting. Next, we analyzed whether ectopic expression of Efnb1 enhances the aggressiveness of Eµ-Myc lymphoma cells in vivo. Since we were not able to culture the Eµ-Myc lymphoma cells from UTX WT or KO mice, we used the Eµ-Myc;p19$^{Arf-/-}$ lymphoma cell line, a well-established mouse B lymphoma cell line commonly used in tumor transplantation studies. Efnb1 cDNA was retro-virally introduced into Eµ-Myc;p19$^{Arf-/-}$ lymphoma cells, and Efnb1-expressing cells were injected into recipient mice to form lymphomas. Compared with lymphoma cells infected with vector control, which developed lymphoma mostly in lymph nodes, lymphoma cells expressing Efnb1 showed enhanced aggressiveness and a more disseminated distribution in mice. In half of the mice bearing the Efnb1-expressing lymphomas, we observed blood in abdomen cavities (Fig. 4a) and visible blood vessels extending towards tumors (Fig. 4b). No mice injected with vector-infected control lymphoma cells showed such phenotypes. In addition, all mice injected with Efnb1-expressing Eµ-Myc; p19$^{Arf-/-}$ cells showed skull bulges due to lymphoma infiltration, which is also frequently seen in Eµ-Myc;UTX-knockout mice. All mice bearing Efnb1-expressing lymphomas showed blood infusion under skull (Fig. 4c), again suggestive of the aggressive nature of Efnb1-expressing lymphomas. In contrast, only 2 out of 9 mice injected with vector control Eµ-Myc;p19$^{Arf-/-}$ cells developed brain bulges, neither of which showed blood infusion. Lastly, in 7 out of 9 mice injected with Efnb1-expressing Eµ-Myc; p19$^{Arf-/-}$ cells, huge lymphoma formed at kidneys (Fig. 4d), whereas only 1 out 9 in the control group showed such phenotype. Taken together, these data showed that overexpressed Efnb1 could recapitulate the dissemination and blood vessel formation phenotype observed in UTX-knockout lymphomas (Fig. 4e). Moreover, analysis of the human lymphoma dataset also showed that UTX-low lymphomas tend to express high levels of Efnb1 (Fig. 4f, g).

**UTX deficiency sensitizes lymphoma cells to Cytarabine**. Given the inferior survival of UTX-low lymphoma patients, we next asked whether potential weakness associated with UTX loss could render lymphoma cells sensitive to certain anticancer drugs, thereby possibly offering better treatment outcome. A recent study of multiple myeloma showed UTX deficiency confers sensitivity to EZH2 inhibitor[26]. Using a previously established experimental platform[27], we analyzed whether UTX knockdown affects the cellular sensitivities towards 24 other types of anticancer drugs in Eµ-Myc;p19$^{Arf-/-}$ lymphoma cells (Fig. 5a, b). Cytarabine (also known as Ara-C), a nucleoside analog that are used primarily in leukemia and occasionally in lymphomas, showed the most significant sensitivity change upon UTX knockdown. UTX-deficient Eµ-Myc;p19$^{Arf-/-}$ lymphoma cells displayed significantly enhanced sensitivity towards cytarabine (Fig. 5c). Interestingly, a recent study of human leukemia suggested that deficiency in EZH2, a histone H3K27 methyl-transferase, caused resistance to cytarabine[28]. This is consistent with our finding that deficiency in UTX, a histone H3K27 demethylase, caused sensitivity to cytarabine. When tested in vivo, cytarabine showed significant activity towards UTX-knockdown lymphomas. For mice bearing UTX-depleted tumors, only 2 out of 20 exhibited palpable tumor 12 days after cytarabine treatment (Fig. 5d). In contrast, 4 out of 10 of mice injected with control lymphoma cells still exhibited palpable tumors at the same time point after drug treatment. Moreover, cytarabine treatment extended overall survival in mice bearing UTX-depleted lymphomas, but not in those bearing control lymphomas (Fig. 5e–g). These in vivo results suggest that UTX-deficient tumors may benefit from cytarabine treatment.

**Discussion**
In summary, our data demonstrated several important aspects of the UTX gene during tumorigenesis. First, the use of tissue-specific UTX-knockout mice allowed us to ask whether UTX is

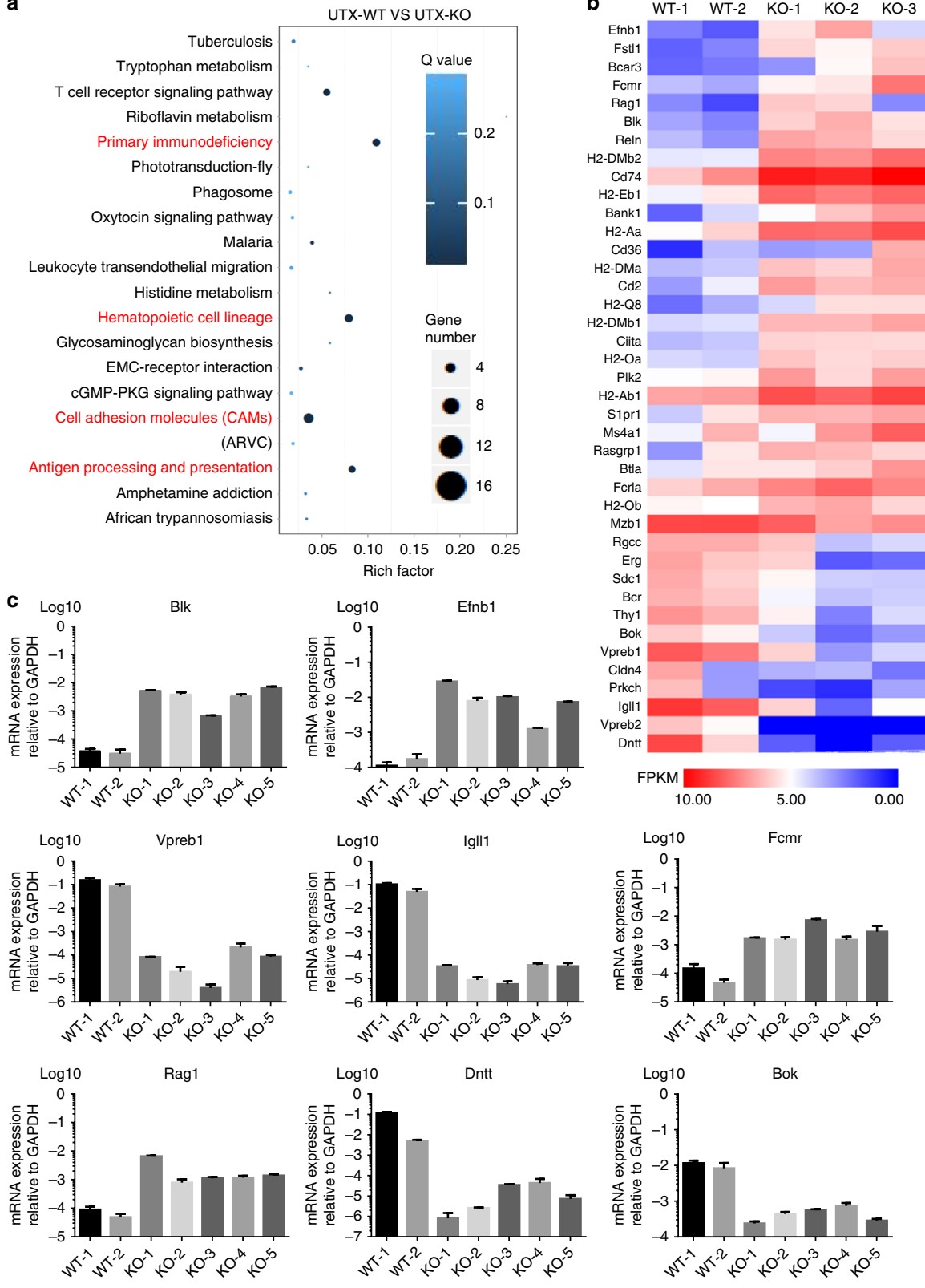

**Fig. 3** Genes involved in B cell biology and cell adhesion are differentially expressed in UTX KO lymphomas. **a** Top 20 of enriched pathway of differentially expressed genes (DEGs) in UTX KO lymphomas. RichFactor is the ratio of differentially expressed gene numbers annotated in this pathway term to all gene numbers annotated in this pathway term. Qvalue is corrected p-value ranging from 0–1, and less Qvalue means greater intensiveness. **b** Representative DEGs involved in B cell biology and tumor development. FPKM method is used in calculated expression level. **c** QPCR confirmation of DEGs' expression in UTX WT and KO lymphomas. Shown are mean ± SEM from three independent experiments

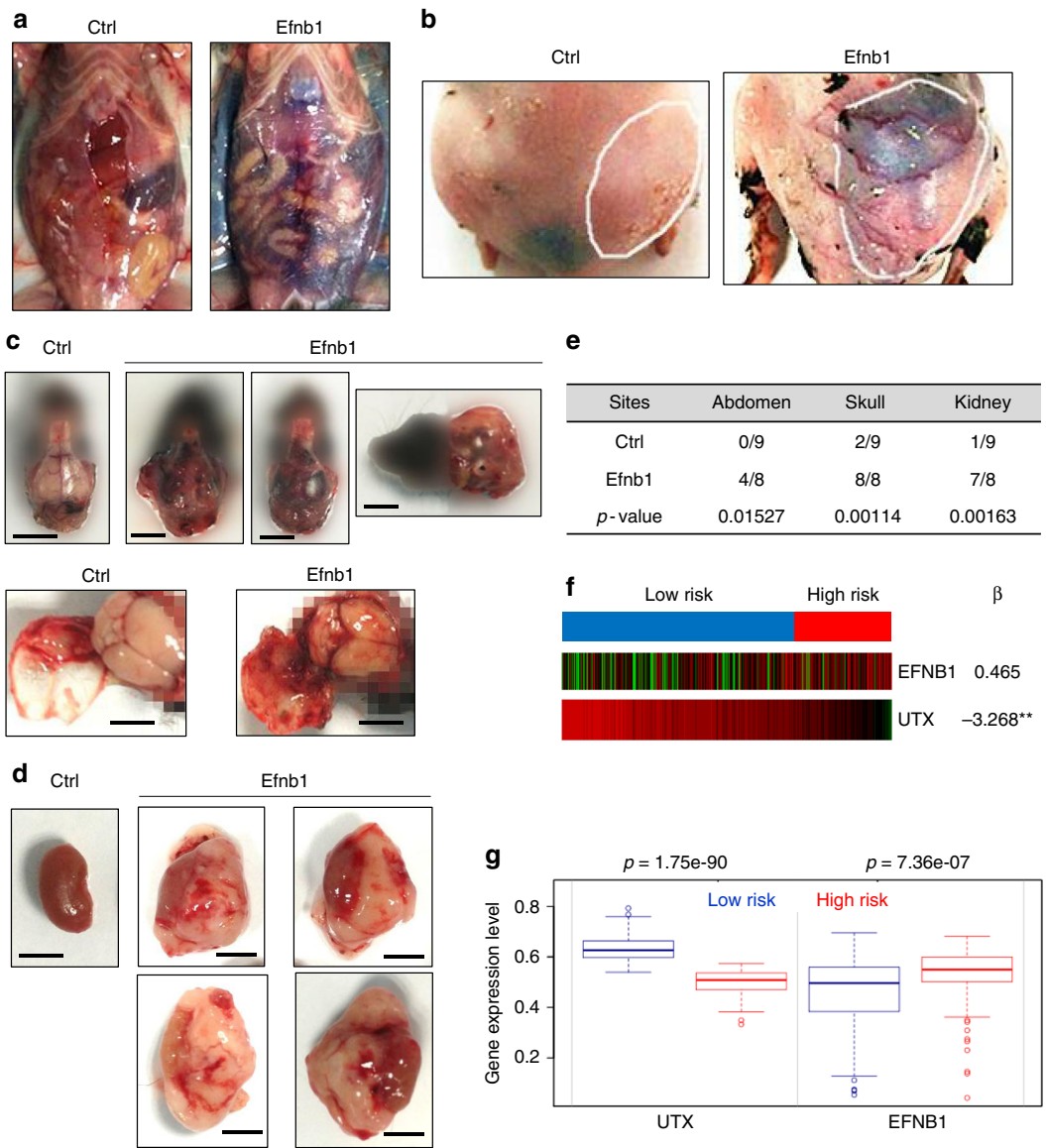

**Fig. 4** Ectopic expression of Efnb1 in Eμ-Myc;p19$^{Arf−/−}$ cells promotes tumor dissemination and blood vessel formation in vivo. **a–d** Representative dissection images of mice injected with Efnb1-expressing lymphoma cells or control cells. **a** Abdomen cavities. **b** Blood vessel formation around the tumor. **c** Infiltration of lymphoma into brain. Bar, 5 mm. **d** Infiltration of lymphoma into kidney. Bar, 5 mm. **e** Summary of lymphoma infiltration or dissemination sites. Ctrl, $n = 9$, Efnb1, $n = 8$. The p-values are determined by the chi-squared test. **f** The pairwise expression of UTX and Efnb1 from dataset Lenz Staudt Lymphoma GSE10846. $N = 420$. Prog.Idx. Prognosis Index. Corresponding beta coefficients from the Cox fitting is shown. Two stars (**) marks genes whose fitting p-value < 0.01, one star (*) marks genes whose fitting p-value < 0.05, and no stars for genes whose p-value is > 0.1, log-rank test. **g** The expression level of UTX and Efnb1 in high risk and low risk groups

indeed an EXITS gene. If UTX does not escape from X-inactivation, we would expect all Eμ-Myc;UTX$^{+/−}$ single knockout mice to possess numerous tumor-generating cells, in which one copy of UTX is deleted, the other copy silenced through X-inactivation. This would result in complete penetrance of lymphoma formation in UTX single knockout mice. However, in our study we observed that only half of Eμ-Myc;UTX$^{+/−}$ mice developed lymphomas. This strongly supports the notion that UTX escapes from X-inactivation, and that UTX is indeed an EXITS gene. Such a finding could help understand the sex-difference in cancer rate observed in humans.

According to the COSMIC database, UTX is frequently mutated, deleted or underexpressed in human cancers. However, as a candidate EXITS gene, the assessment of UTX is further complicated by the fact that its paralog UTY is located on Y chromosome. Although UTY is previously reported to be devoid of demethylase activity, a recent biochemical study questioned this notion[29]. Moreover, Shpargel et al. showed UTX$^{−/−}$ mice were embryonic lethal, whereas UTX$^{−/Y}$ mice developed to term, and a portion of UTX$^{−/Y}$ mice reached adulthood[30]. This indicates that during development, UTY could compensate for UTX. To understand the importance of UTX as an X-chromosome-encoded tumor suppressor, it is necessary to know whether UTY significantly compensates for UTX during tumorigenesis. Our use of male and female UTX conditional knockout mice potentially allowed us to address this question. The results suggested that UTY provides a partial, but rather weak compensation for UTX during tumorigenesis. However, given that other factors may also influence tumorigenesis in male and female mice, the most stringent way to answer this question is to study it in UTX;UTY

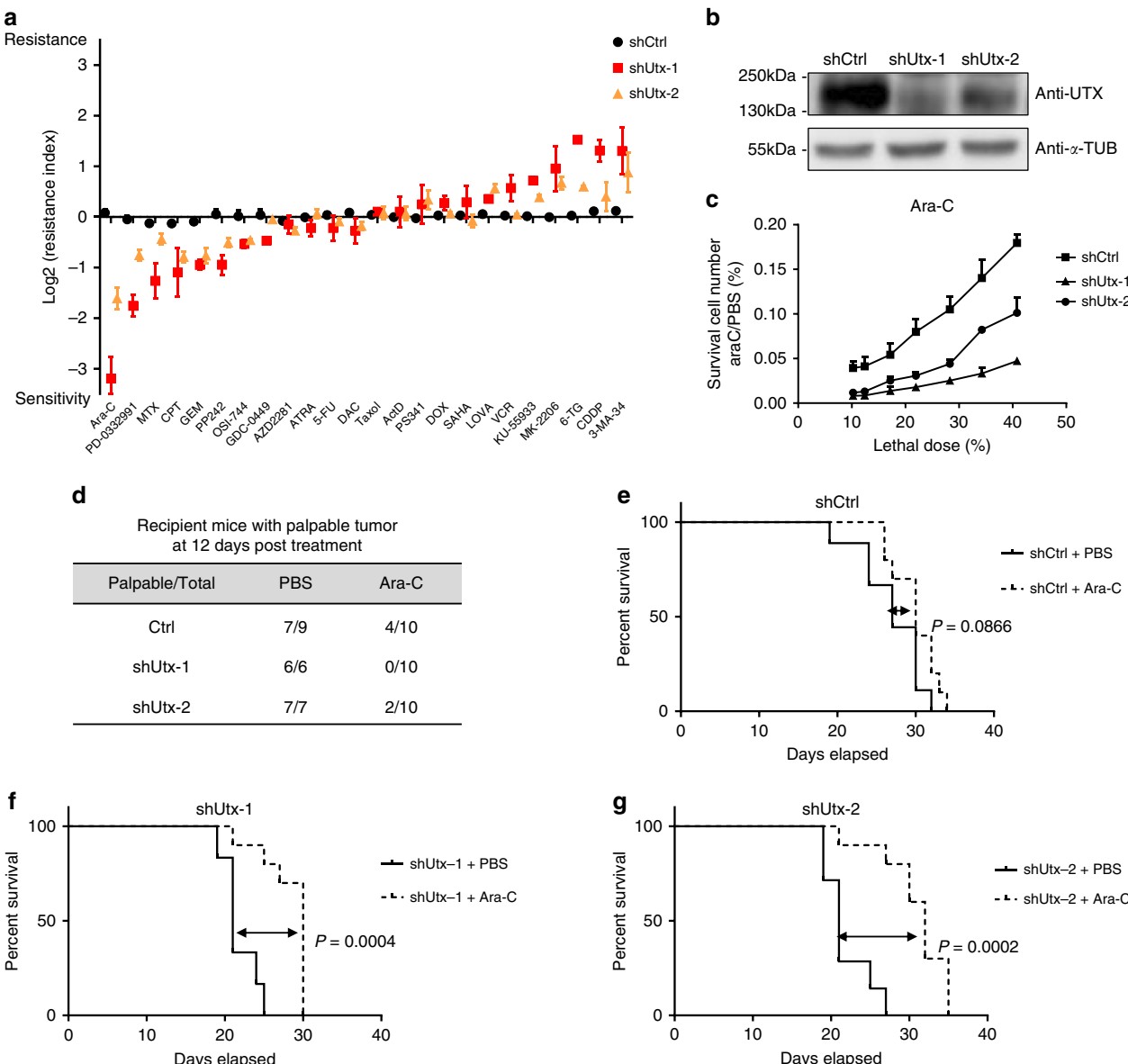

**Fig. 5** UTX deficiency renders lymphoma cells hypersensitive to Cytarabine. **a** UTX-knockdown cells showed altered sensitivity towards certain drugs. Shown are mean ± SEM from three independent experiments. **b** A western blot showing knockdown efficiency of UTX. **c** Upon knockdown of UTX, Eμ-Myc; p19$^{Arf−/−}$ lymphoma cells become more sensitive to Ara-C treatment. Shown are mean ± SD from three independent experiments. **d** Number of mice with palpable lymphomas 12 days post treatment. **e**–**g** Kaplan–Meier curves indicating the survival of mice injected with control or UTX-knockdown lymphoma cells, with or without Ara-C treatment. Statistical significance was determined using log-rank test

double conditional knockout mice. Nonetheless, our results, combined with the finding that UTY is rarely mutated or deleted in cancer (http://cancer.sanger.ac.uk/cosmic/gene/analysis?ln=kdm6b), suggest that deficiency of the only copy of UTX in males is expected to sufficiently allow for tumorigenesis, and the presence of UTY on Y chromosome would not significantly impede tumor formation caused by UTX loss. This conclusion also further validates the importance of UTX as an EXITS gene.

Previous ex vivo mouse leukemia models using UTX$^{−/Y}$ or UTX-knockdown cells demonstrated UTX loss accelerates initiation of malignancy. However, UTX's potential impacts on tumor progression remain unknown. Interestingly, our results showed that UTX loss not only promoted lymphomagenesis, it also caused enhanced aggressiveness of the resulting tumor. Importantly, in human lymphoma patients, low UTX expression

in tumor is also associated with poor survival. Through transcriptome analysis, we found Efnb1 is overexpressed in UTX-knockout tumor, and subsequent experiments demonstrated that Efnb1 overexpression in lymphoma cells indeed recapitulated the enhanced dissemination and blood vessel formation phenotypes in UTX-knockout mice. Our results showed that Efnb1 is sufficient to cause the aggressive phenotypes seen in UTX-knockout tumors. Whether it is necessary for UTX-knockout tumors to produce such phenotypes remains to be tested. In addition, ectopic expression of Efnb1 in Eμ-Myc;p19$^{Arf−/−}$ lymphoma cells could also provide a tractable system to study brain metastasis of lymphomas, which is an important cause of death for lymphoma patients.

Lastly, from a precision medicine perspective, we observed that UTX depletion sensitized lymphoma cells to cytarabine. Interestingly, this observation is consistent with a recent finding that

loss of Ezh2, an enzyme with opposing function to UTX, confers resistance to cytarabine[28]. Moreover, recent clinical trial results also showed that using high-dose cytatabine may be benifical in treating aggressive B cell lymphomas[31–33]. Of note, the lymphoma patients group analyzed in our study was treated with CHOP-like or R-CHOP-like regimen. Under such situation the UTX-low lymphoma caused poor survival. Therefore, it may be interesting to test whether lymphoma regimens that contain cytarabine, such as HyperCVAD, DHAP, and BEAM could potentially enhance treatment outcome of UTX-deficient lymphomas.

## Methods

**Cell lines, chemicals, antibodies, and constructs.** 293T cells was purchased from ATCC and was cultured in 90% DMEM, 10% fetal bovine serum, supplemented with 100 U/ml penicillin and streptomycin,. Eμ-Myc;p19$^{Arf−/−}$ mouse B-cell lymphoma cell was a kind gift from Dr. Michael Hemann at MIT, and was cultured in 45%DMEM, 45%IMDM, 10% fetal bovine serum, supplemented with 100 U/ml penicillin and streptomycin, and 25 μM β-mercaptoethanol[27]. All cell cultures were tested every 2 weeks for mycoplasma contamination. There were no signs of mycoplasma contamination for all cell lines. Cell line authentication was performed via short tandem repeat profiling. Chemicals were obtained from Selleck, Sigma, Tocris, or Calbiochem. Antibodies against UTX (A302-374A, Bethyl Lab, USA, 1:1000) and Tubulin (T-5168, Sigma, USA, 1:5000) were used for western blot analysis (Fig. 5b and Supplementary Figure 6). Antibodies against B220 (RA3-62B, BD Pharmingen, USA), CD19 (ID3, BD Pharmingen, USA), CD43 (S7, BD Pharmingen, USA), IgM (II/41, BD Pharmingen, USA), CD5(53-7.3, BD Pharmingen, USA), CD23(3B4, BD Pharmingen, USA) were used in FACS analysis of lymphoma cells at a concentration of 1:200.

shRNAs are cloned into retroviral MLP and MLS vectors using the following procedures[27]. RNAi sequences against UTX were selected using the Siscales program (http://gesteland.genetics.utah.edu/siRNA_scales/). 97-mer DNA oligos containing the RNAi sequences were designed using the RNAi central website (http://katahdin.cshl.org/homepage/siRNA/RNAi.cgi?type=shRNA). The oligos were synthesized and PCR-amplified with primers (XhoI-primer: 5′-CAGAAGGCTCGAGAAGGTATATTGCTGTTGACAGTGAGCG-3′ and EcoRI-primer: 5′-CTAAAGTAGCCCCTTGAATTCCGAGGCAGTAGGCA-3′). PCR products were digested with EcoRI and XhoI enzymes, and ligated to MLP or MLS vectors that were predigested with EcoRI and XhoI.

Coding sequence of Efnb1 was amplified from lymphoma cDNA. qPCR primer sequences and shRNA sequences are provided in Supplementary Tables 4 and 5.

**Human lymphoma datasets analysis.** SurvExpress (http://bioinformatica.mty.itesm.mx/SurvExpress) was used to provide survival analysis and risk assessment[17]. The database of Lenz Staudt Lymphoma GSE10846 was chosen, which contained larger sample set ($n = 420$) with gender information. SurvExpress performs risk grouping through optimization algorithm. This program tests whether certain gene or gene combination can be used as biomarkers that are able to separate risk groups characterized by differences in gene expression. We used UTX as biomarkers/variables in Fig. 1a, b, and UTX and EFNB1 combination as biomarkers/variables in Fig. 4f–g.

**Generation of mice and tumor monitoring.** All mice were housed in a specific pathogen-free environment at the Shanghai Institute of Biochemistry and Cell Biology and treated in strict accordance with protocols, which were approved by the Institutional Animal Care and Use Committee of the Shanghai Institutes for Biological Sciences, Chinese Academy of Sciences.

Eμ-Myc mice were purchased from the Jackson laboratory (Jackson no. 002728). UTX$^{f/f}$;CD19-Cre$^{+/+}$ mice were provided by Charlie Degui Chen Lab. The Eμ-Myc strain was maintained by mating hemizygous male to a wild-type C57BL/6J female. To generate the Eμ-Myc;UTX$^{f/y}$;CD19-Cre$^{+/−}$ and Eμ-Myc; UTX$^{f/f}$;CD19-Cre$^{+/−}$ mice, we first bred Eμ-Myc;UTX$^{+/y}$ mice with UTX$^{f/f}$ mice to get Eμ-Myc;UTX$^{f/y}$ mice, then bred Eμ-Myc;UTX$^{f/y}$ mice with UTX$^{f/f}$; CD19-Cre$^{+/+}$ mice to generate the Eμ-Myc;UTX$^{f/y}$;CD19-Cre$^{+/−}$ and Eμ-Myc; UTX$^{f/f}$;CD19-Cre$^{+/−}$ mice. We bred Eμ-Myc;UTX$^{+/y}$ mice with UTX$^{f/f}$; CD19-Cre$^{+/+}$ mice to generate the Eμ-Myc;UTX$^{f/+}$;CD19-Cre$^{+/−}$ mice. We bred Eμ-Myc;UTX$^{f/y}$ mice with UTX$^{f/f}$ mice to generate the Eμ-Myc;UTX$^{f/y}$ and Eμ-Myc;UTX$^{f/f}$ mice. All mice were maintained in a C57BL/6 background.

Mice were monitored for tumor formation twice a week by palpation of the brachial, axillary, and cervical lymph nodes. Enlargements of at least 5 mm in the longest diameter were considered "well palpable" and reflect lymphoma formation[34].

**RNA-Seq analysis.** For the transcriptome analysis, we dissected lymphomas from major tumor bulks in mice (both nodel and extra-nodel), meshed them and FACS-sorted lymphoma cells. The lymphomas are mostly composed of lymphoma cells with more than 90% purity. These lymphoma cells are single cells and pass through filter upon centrifugation, whereas most non-tumor cell structures are filtered out. The filtered cells are treated with red cell lysis buffers to further remove red blood cells before FACS sorting. Lymphoma cells are very small, round cells and can be identified through forward scatter and side scatter in FACS. After sorting, their identities were confirmed by further cell surface marker staining.

For the transcriptome analysis, the tumors were collected from littermate UTX WT and KO mice. In UTX WT and KO mice, the B cell developmental stages of lymphomas are different. The tumors in this littermate are: UTX-WT-1 (pre-B), UTX-WT-2 (pro-B), UTX-KO-1,-2 (Immature/mature B), UTX-KO-3 (pre-B), and UTX-KO-4,-5 (Immature/mature B).

RNA-Seq was performed using the BGIseq500 platform (BGI, Wuhan, China, http://www.seq500.com/en/). We first used NOISeq method to screen differentially expressed genes before performing pathway enrichment analysis. For NOISeq method, samples should be firstly grouped so that comparison between every two groups as a control-treatment pairwise can be done later. The provided group information is as following: UTX-WT: UTX-WT-1, UTX-WT-2. UTX-KO: UTX-KO-1, UTX-KO-2, UTX-KO-3. Significantly different expression with an absolute value of $\log_2$Ratio(UTX-KO/UTX-WT) $>= 1$ and a probability value $> 0.8$ were used for further function analysis. Pathway enrichment analysis of DEGs was performed based on KEGG database.

**Lymphoma transplantation and in vivo cytarabine treatment.** Efnb1-expressing virus or vector control, both expressing GFP, were used to infect lymphoma cells. $10^6$ sorted GFP + lymphoma cells in 100 μl of PBS were injected into C57BL/6 females recipient mice to monitor response via tail vein. At time of tumor transplantation, these mice were six weeks old. The numbers of mice used in this experiment were noted in Fig. 4. Mice were monitored daily for survival and sacrificed at Day 20 to assess tumor presentation.

To test in vivo drug sensitivity, control or UTX-knockdown cells were transplanted via tail vein injection. The recipient mice were six-week-old C57BL/6 females and the numbers of mice in this experiment were noted in Fig. 5. For mice injected with UTX-knockdown and control lymphoma cells, PBS or cytarabine (200 mg per kg dose) was i.p. injected into mice for five consecutive days. Mice were monitored for tumor formation at 12 days post of treatment. Mice were monitored daily for survival after drug treatment.

**Analysis of drug sensitivity change caused by UTX deficiency.** To test how shRNA suppression of UTX affects cellular sensitivity to drugs, we used a GFP-based competition experimental system[27]. Briefly, UTX shRNA along with GFP was stably expressed in a portion of Eμ-Myc;p19$^{Arf−/−}$ lymphoma cells, and the mixture of green (UTX-deficient) and white (UTX-proficient) cells were subjected to drug treatment. If UTX knockdown sensitizes cells to a drug, then we would observe a decrease of GFP percentage in drug-treated, surviving population. The GFP percentages with and without drug treatment can be used to quantitatively assess how many fold is resistant the UTX-knockdown cells are compared to UTX proficient cells. The calculation format for relative resistance index[27] is as following: Relative resistance Index = (G2-G2*G1)/(G1-G1*G2). G1 equals the percentage of GFP-positive cells in no-treatment cell population, and G2 equals the percentage of GFP negative cells in drug-treated cell population. For example, if after cytarabine treatment, the percentage of GFP-positive, UTX-knockdown cells decreases from 40% to 20%, then the relative resistance index can be calculated as (0.2−0.4*0.2)/(0.4−0.4*0.2) = 0.375. This means UTX-knocked down cells is 0.375 fold as resistant to cytarabine compared to UTX-expressing cells.

Drug used in this study is listed in Supplementary Table 6.

**Statistical evaluation.** Mouse survival data were plotted with the Kaplan–Meier population-event time course format and compared using the log-rank (Mantel–Cox) test.

**Data availability.** RNA-Seq data of this study have been deposited in NCBI Sequence Read Archive (SRA) with accession codes SRP135245. All data that support the finding of this study are available from the corresponding author upon reasonable request.

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

## Acknowledgements

This work was supported by the major scientific research project from the Ministry of Science and Technology (Grant No. 2017YFA0504503, 2013CB910404), the Strategic Priority Research Program of Chinese Academy of Sciences, Grant No. XDB19000000, and Natural Science Foundation of the People's Republic of China (Grant No. 31371418). We thank Prof. Hongbin Ji for discussions and helpful comments, Jingyao Zhao for assistance on FACS analysis, Zhuang Wei for assistance on histopathological analysis, and Animal Core Facility and Core Facility for Cell Biology at SIBCB.

## Author Contributions

H. J. and X.-X. L. designed the experiments. X.-X. L. conducted the experiments. Y.-L. Z. and M.-X. L. maintained the mice. UTX^{f/f} mice were generated in C.D. C.'s lab and L.Z. crossed and maintained CD19-CRE;UTX^{f/f} mice. H. J. and X.-X. L. analyzed the data and wrote the paper. All authors discussed the results and commented on the manuscript. H.J. supervised the study.

## Additional information

**Competing interests:** The authors declare no competing interests.

