## [Peer Review File · Nature Communications]

Reviewers' comments:

Reviewer #1 (Remarks to the Author):

Li and colleagues present functional data in a conditional knock-out mouse model that provides an elegant demonstration of sex-biased tumor suppressor function for UTX/KDM6A, a gene that escape X-inactivation in female cells. Using an Eu-MYC driven B cell leukemia/lymphoma model, they demonstrate that UTX is a dosage sensitive tumor suppressor and may be responsible for some of the increased incidence and possibly aggressiveness of male cancers that harbor UTX mutations. This is a novel finding. UTX has been shown previously to have male mutation bias in cancer. And UTX tumor suppressor function in T-ALL has been shown previously using shRNAs and knock-out animals. However, this is the first report I am aware of that specifically compares males with 0 or 1 copy and females with 0, 1, or 2 copies of UTX in a germline oncogene-driven cancer model. These data are an important experimental demonstration of the theory that females have relative protection from tumorigenesis by having two expressed alleles of these so-called 'EXITS' genes, also in another lineage.

Comments:

The risk modeling in Figure 1AB is not well explained. It is unclear what variables were used to build the model, and whether gender and UTX level are independent variables? What other known risk factors in patients were considered, and is UTX level independently prognostic?

In the UTX^{+/-} females that get lymphoma, do they lose or disable the other copy of UTX? This could be examined at DNA, RNA, and/or protein level. Also, what is the cell surface/developmental stage phenotype of the UTX^{+/-} female lymphomas? If they are more like the KO tumors, then this may signal they have lost the other allele of UTX.

Gene expression profiling of wild-type vs KO lymphomas is not very surprising if these were randomly chosen tumors because the authors showed that KO lymphomas are of a different developmental stage and many of the identified genes are known to be differentially expressed during B cell maturation. To put the question another way, what was the developmental stage of the tumors profiled in Figure 3B and 3C? Were they similar or different? Alternatively, could the authors return to their human lymphoma gene expression data and determine if similar expression patterns are found in UTX mutant compared to non-mutant tumors?

The transition to the Arf null cell experiments is not entirely logical. Why not first knock out UTX in the Arf null cells and show a change in aggressivity phenotype and level of Efnb1? Then could show the Efnb1 overexpression phenotype as presented now. This would then allow performance of the necessary/sufficient experiments to test if knocking down Efnb1 in UTX KO cells can reverse the UTX KO phenotype.

Similar to comments for Figure 1 risk modeling, need better explanation of experiments done for Figure 4F. Need statistical test for anti-correlation of UTX and EFNB1 levels. Also in the Efnb1 mouse model, need a statistical test to support statement that EFNB1 overexpressing cells have a different disease phenotype.

Minor comments:

Figure 2D (spleen pathology) is not particularly informative and too small/too low power. Higher power and/or IHC for specific lymphoid cell markers may help? And if going to use the term "angiogenesis," including in abstract, would need to at least perform pathology stains for vessels in some tissue.

Increased sensitivity of UTX null cells is very interesting in light of clinical trial data that high-dose cytarabine may be particularly active in aggressive B cell lymphomas, such as mantle cell lymphoma. May consider citing some of that literature (e.g., Hermine et al, Lancet 2016, PMID:

27313086).

Why is the most upregulated gene in MYC KO tumors Eif2s3y (Table S2)? According to the text, these are all supposed to be female animals, and Eif2s3y is a Y chromosome gene.

Reviewer #2 (Remarks to the Author):

This is an interesting study in which the authors study how the escape of X inactivation of the X-linked tumor suppressor gene Utx, contributes to differences in tumor incidence between males and females. I believe this is the first study that addresses this phenomena using in vivo mouse models. Nevertheless, there are major issues that should be addressed.

Major comments

1. The authors state that CD19-CRE Utx fl/fl mice are not developing any tumors by day 200. I believe that these animals should be followed longer to see if they develop any malignancy. At least follow up for 1 year should be presented.
2. Figure 1 E. In this figure, the authors compare male and female mice lacking UTX. However, there is already a considerable difference in tumor initiation between the Eu-MYC male and female mice. How does this influence the current data and the conclusions?
3. the authors claim that loss of UTX results in advanced lymphoma, including brain metastasis, which caused skull bulges, and huge thymic/mediastinal tumors. However, and as also mentioned by the authors, i believe that the data rather shows that loss of Utx causes a swith in malignancy rather than metastasis of late-stage lymphoma. As characterised by the authors, the tumors might reflect a different type of B-cell leukemia/lymphoma. I would rather say that loss of UTX results in more leukaemic features (instead of defining it as metastasis).
4. The authors mention the occurrence of huge thymic tumors upon loss of UTX. Did the authors analyse the thymic tumor material to check if these tumor cells were from B cell origin?
5. the authors performed transcriptomics on the mouse tumors obtained in this study. How did they perform these experiments? Did they sort tumor cells? From what tissue?
6. Besides transcriptomics, at least some mechanistic insights should be provided in relation to epigenetic deregulation upon loss of UTX. UTX CHIPseq and H3K27me3 ChIPseq should be performed on WT and UTX KO tumors, to correlate with the observed changes in gene expression.
7. theEfnb1 experiment is largely correlative. this setup is not testing whether the UTX phenotype is partly due to Efnb1. For this, one should do a rescue experiment

Minor point

In the second part of the result section. The authors state "This strongly supported the theory proposed by Dunford et al, and showed that UTX is a key candidate for the 'escape from X-inactivation tumor-suppressor' (EXITS) genes". However, this concept was first introduced by an earlier paper of Van der Meulen et al. (Blood 2015; reference 11)

Reviewer #3 (Remarks to the Author):

Li et al report that UTX has a role as an X-inactivation tumor suppressor and accelerates tumor growth in a myc-associated lymphoma model. They suggest that UTX deletion promotes malignant cell dissemination and is associated with increased angiogenesis, and that this is associated with increased expression of Efnb1. They also suggest that UTX deficiency renders the tumor cells more sensitive to cytarabine therapy.

Specific comments -

1. X-chromosome gain is seen with similar frequency to X-chromosome loss in B-cell lymphoma and when present is typically only seen in a small percentage of cells. To confirm the relevance of UTX loss in the pathogenesis of lymphoma particularly because deletion of UTX in B-cells alone did not induce lymphoma, the authors should show decreased expression of UTX in most malignant cells in various lymphomas.
2. The authors need to address the clinical relevance of the tumor model. They use a myc-driven model of lymphoma (which is more similar to a Burkitt lymphoma than mature lymphomas), yet they then study the prognostic value in a dataset of diffuse large B-cell lymphoma rather than Burkitt lymphoma. Because the UTX^{-/-} Eu-Myc mice have a leukemic phenotype (and UTX loss has been associated with ALL), the authors should also discuss why they did not use a leukemia patient database for clinical correlations.
3. The evidence that UTX deficiency is associated with an increased propensity to disseminate or increased angiogenesis is not convincing. Increased disease in the brain or other organs, and increased "blood infusion" in organs, seems to be more extensive disease due to the more aggressive growth pattern that has been allowed to progress rather than a functional change in the tumor cells. It is likely that control cells would result in a similar picture if allowed to progress for a few more days. The authors need to do additional studies to show that the cells migrate and adhere differently and studies to show increased new vessel formation in the presence of UTX-deficient cells compared to controls.
4. The authors need to reconcile the finding that UTX-deficient cells are associated with increased sensitivity to cytarabine with their finding that UTX loss is associated with a poor prognosis. Increased sensitivity to treatment should result in more patients been salvaged with standard cytarabine-containing chemotherapy.

Minor comment -

Ezponda et al (Cell Rep 2017) recently published that UTX loss promotes a more aggressive phenotype in myeloma (another mature B-cell process) and sensitizes cells to EZH2 inhibition. The authors should reference this publication and discuss whether UTX deficient cells in their model are more sensitive to EZH2 inhibitors.

Response to questions

Reviewer #1 (Remarks to the Author):

Li and colleagues present functional data in a conditional knock-out mouse model that provides an elegant demonstration of sex-biased tumor suppressor function for UTX/KDM6A, a gene that escape X-inactivation in female cells. Using an E μ -Myc driven B cell leukemia/lymphoma model, they demonstrate that UTX is a dosage sensitive tumor suppressor and may be responsible for some of the increased incidence and possibly aggressiveness of male cancers that harbor UTX mutations. This is a novel finding. UTX has been shown previously to have male mutation bias in cancer. And UTX tumor suppressor function in T-ALL has been shown previously using shRNAs and knock-out animals. However, this is the first report I am aware of that specifically compares males with 0 or 1 copy and females with 0, 1, or 2 copies of UTX in a germline oncogene-driven cancer model. These data are an important experimental demonstration of the theory that females have relative protection from tumorigenesis by having two expressed alleles of these so-called 'EXITS' genes, also in another lineage.

Comments:

1. The risk modeling in Figure 1AB is not well explained. It is unclear what variables were used to build the model, and whether gender and UTX level are independent variables? What other known risk factors in patients were considered, and is UTX level independently prognostic?

Response:

The SurvExpress analysis (Aguirre-Gamboa et al, 2013) was used to test whether candidate gene or gene combination can be used as biomarkers that are able to separate risk groups characterized by differences in gene expression. In Figure 1A, UTX expression level was used as the single variable to analyze their predictive power on patient survival. The low risk group is the group of patients in which UTX expression was high, and the high risk group expresses low level of UTX. The purpose of Figure 1B is to show the distribution of male and female patients according to UTX expression, however gender itself was not used to generate the survival curve analysis.

As suggested by the reviewer, we added additional risk modeling in male and female respectively as supplementary figures. When using gender as the variable, no significant difference has been seen between male and female patients on overall survival (Supplementary Figure 5a).

When using UTX level as the variable, significant difference has been seen in both male and female patients on overall survival (Supplementary Figure 5b-c).

These data combined with Figure 1A-B show that UTX level but not gender is independently prognostic in human lymphomas. We added such information to the revised manuscript (Page 7)

Supplementary Figure 5

2. In the UTX^{+/-} females that get lymphoma, do they lose or disable the other copy of UTX? This could be examined at DNA, RNA, and/or protein level. Also, what is the cell surface/developmental stage phenotype of the UTX^{+/-} female lymphomas? If they are more like the KO tumors, then this may signal they have lost the other allele of UTX.

Response:

The tumorigenesis curves of UTX^{+/-} vs UTX^{-/-} mice suggest that the single copy of UTX does present a barrier to tumorigenesis, and it is possible that in those cases where the UTX^{+/-} mice do develop tumor, the remaining UTX is lost. Alternatively, some other cancer genes or cancer-promoting pathways may be altered in these mice to allow for tumorigenesis. We are able to analyze the UTX status in three UTX^{+/-} lymphoma lysates through Western Blot. All three retained the expression of UTX (shown below). This suggests that in these UTX^{+/-} cells, other tumor-promoting events may occur to enable tumorigenesis in the presence of UTX. However we do not have samples for all UTX^{+/-} lymphomas, and we can not rule out the possibility that in a portion of UTX^{+/-} mice, cells progressed to tumors through loss of the remaining UTX copy.

The cell surface marker analysis of the UTX^{+/-} female lymphomas indicated they originated from late stage of B cell development. They are indeed more like the KO tumors. Since we do not have enough experimental evidence to make the conclusion on what percentage of UTX^{+/-} female lymphomas have lost the other allele of UTX, we did not show the UTX^{+/-} results in the manuscripts.

Western Blot of the expression level of UTX in lymphomas from UTX^{+/+}, ^{+/-} and ^{-/-} mice

3. Gene expression profiling of wild-type vs KO lymphomas is not very surprising if these were randomly chosen tumors because the authors showed that KO lymphomas are of a different developmental stage and many of the identified genes are known to be differentially expressed during B cell maturation. To put the question another way, what was the developmental stage of the tumors profiled in Figure 3B and 3C? Were they similar or different? Alternatively, could the authors return to their human lymphoma gene expression data and determine if similar expression patterns are found in UTX mutant compared to non-mutant tumors?

Response:

The UTX KO tumors mostly display late stage B cell markers, whereas UTX wild-type

tumors are mostly early stage. This indicates that UTX probably influences epigenetic status in B cells that results in such phenomenon. Therefore, when we conducted the transcriptome analysis in UTX WT and KO tumors, we decided not to choose tumors that are matching B cell developmental stage, as it may bring in interference. Instead, we used tumor samples from littermate UTX WT and KO mice for the transcriptome analysis. In UTX WT and KO mice, the B cell developmental stages of lymphomas are different. The tumors in this littermate are: UTX-WT-1 (pre-B), UTX-WT-2 (pro-B), UTX-KO-1,-2 (Immature/mature B), UTX-KO-3 (pre-B). We added such information to the manuscript (Page 16) as suggested by the reviewer.

Results of the transcriptome profile analysis showed differential expression of many B cell mature genes in UTX WT and KO tumors, and it further confirmed that UTX KO lymphomas are originated from late stage. Although such results reflect the development stage of these tumors, the transcriptome analysis also showed differential expression of cell adhesion genes in UTX WT and KO tumors, which led to discovery of deregulated genes in the UTX KO tumors that are potentially responsible for the aggressive phenotypes of UTX KO tumors, such as *Efnb1*.

In addition, although widely used in lymphoma research, the E μ -Myc lymphoma model was known to produce mostly pre-B and pro-B tumors, which are very different from the human case. In humans, lymphomas are mostly originated from late stage B cells. The E μ -Myc;UTX KO mice therefore provide a lymphoma model that's more close to the human disease setting.

4. The transition to the Arf null cell experiments is not entirely logical. Why not first knock out UTX in the Arf null cells and show a change in aggressivity phenotype and level of Efnb1? Then could show the Efnb1 overexpression phenotype as presented now. This would then allow performance of the necessary/sufficient experiments to test if knocking down Efnb1 in UTX KO cells can reverse the UTX KO phenotype.

Response:

In our experiment, we observed increased *Efnb1* expression in UTX KO lymphomas. Since the UTX KO or WT lymphomas can not be grown in tissue culture, we used the E μ -Myc;Arf null cell line and showed that *Efnb1* overexpression alone can mimic the enhanced aggressiveness on UTX KO tumors, such as brain metastasis.

We have tried many methods to generate a UTX KO E μ -Myc;Arf cells, but ultimately couldn't do it. The E μ -Myc;Arf cells are small, floating cells, and is very hard to transfect. It is also very poorly infected by lentivirus. We've tried many conditions, however we cannot introduce Cas9 into these cells by transfection or lentiviral transduction. These cells are receptive to retrovirus infection, but the size of the Cas9 gene exceeds the packing limit of retroviral system. Therefore, although *Efnb1* is sufficient for promoting tumor aggressiveness, we cannot generate UTX KO E μ -Myc;Arf cells to test whether *Efnb1* is necessary for UTX KO's aggressive phenotype. In light of this, we made relevant changes in the manuscript. (Page 2, 9,12)

5. Similar to comments for Figure 1 risk modeling, need better explanation of experiments done for Figure 4F. Need statistical test for anti-correlation of UTX and EFNB1 levels. Also in the Efnb1 mouse model, need a statistical test to support statement that EFNB1 overexpressing cells have a different disease phenotype.

Response:

Firstly, we add additional Kaplan-Meier data by male and female group respectively according to reviewer's suggestion (Supplementary Fig 6).

The SurvExpress analysis (Aguirre-Gamboa et al, 2013) was used to test whether candidate gene or gene combination can be used as biomarkers that are able to separate risk groups characterized by differences in gene expression. In Figure 4F, UTX and EFNB1 expression levels were used as the variable to analyze their predictive power on patient survival. For this, we performed risk grouping analysis in SurvExpress by prognostic index median and Cox fitting. The Corresponding beta coefficients from the Cox fitting and p-value for UTX and EFNB1 are shown (Fig 4f 4g). (Page 9-10)

We found the UTX-low;EFNB1-high status (Fig 4h, Supplementary Fig 6) is able to separate risk groups characterized by expression levels of these genes. Moreover, the p-value of the risk group separation, the concordance index, and the significance of the coefficients using UTX-EFNB1 were slightly better than that using UTX alone. Therefore, the results show that for predicting survival of human DLBCL, the biomarker of UTX-low and ENFB1-high together will work better than UTX alone. (Page 9-10)

Finally, as suggested, we add chi-squared test to support statement on EFNB1 mouse phenotype (Fig 4e).

Supplementary Figure 6

Risk grouping by UTX and Efnb1 expression data.

Risk grouping by UTX and Efnb1 expression data.

Minor comments:

6. Figure 2D (spleen pathology) is not particularly informative and too small/too low power. Higher power and/or IHC for specific lymphoid cell markers may help? And if going to use the term “angiogenesis,” including in abstract, would need to at least perform pathology stains for vessels in some tissue.

Response:

We supplemented higher power figures in Supplementary Figure 3 in revised version.

We found major blood vessels on the surface and within the lymphomas, but we can't find smaller vessels in HE staining as show in Supplementary Figure 3b. Given that the lymphomas are a semi-solid mass and there are no rigid structures within lymphomas that

strongly restrict diffusion of oxygen and nutrients, it may not be necessary to further vascularize in the lymphomas; the nutrients and oxygen brought by major vessels is easily transported into the inner of lymphomas due to the specific histology of lymphomas. In contrast, lymphomas in UTX^{+/+} mice did not develop such massive blood vessels. Such a phenotype demonstrates that UTX^{-/-} lymphomas are more aggressive. However, lack of small blood vessels in HE staining may not strictly adhere to the definition of angiogenesis, and we therefore avoided using “angiogenesis” in describing our results in the revised version.

Supplementary Figure 3

7. Increased sensitivity of UTX null cells is very interesting in light of clinical trial data that high-dose cytarabine may be particularly active in aggressive B cell lymphomas, such as mantle cell lymphoma. May consider citing some of that literature (e.g., Hermine et al, Lancet 2016, PMID: 27313086).

Response:

We thank the reviewer for this suggestion as it may increase the potential clinical relevance of our finding. We cited the following literatures on cytarabine in lymphoma clinical trials (Page 12-13). These literatures suggest the potential usefulness of cytarabine in treating aggressive lymphomas.

- 1 Hermine, O., et al. (2016). "Addition of high-dose cytarabine to immunochemotherapy before autologous stem-cell transplantation in patients aged 65 years or younger with mantle cell lymphoma (MCL Younger): a randomised, open-label, phase 3 trial of the European Mantle Cell Lymphoma Network." *Lancet* **388**(10044): 565-575.
- 2 Umino, K., et al. (2017). "High-Dose Methotrexate and Cytarabine-Based Multi-Agent Chemotherapy (Modified Bonn Protocol) for Systemic Lymphoma with CNS Involvement." *Acta Haematol* **137**(2): 93-99.
- 3 Gonzalez-Barca, E., et al. (2016). "Central nervous system prophylaxis with intrathecal liposomal cytarabine in a subset of high-risk patients with diffuse large B-cell lymphoma receiving first line systemic therapy in a prospective trial." *Ann Hematol* **95**(6): 893-899.

8. Why is the most upregulated gene in MYC KO tumors Eif2s3y (Table S2)? According to the text, these are all supposed to be female animals, and Eif2s3y is a Y chromosome gene.

Response:

This is a mistake in data organization. We apologize for such a mistake and made corrections in the table. (Supplemental table 2)

Reviewer #2 (Remarks to the Author):

This is an interesting study in which the authors study how the escape of X inactivation of the X-linked tumor suppressor gene *Utx*, contributes to differences in tumor incidence between males and females. I believe this is the first study that addresses this phenomena using in vivo mouse models. Nevertheless, there are major issues that should be addressed.

Major comments

1. The authors state that CD19-CRE *Utx* fl/fl mice are not developing any tumors by day 200. I believe that these animals should be followed longer to see if they develop any malignancy. At least follow up for 1 year should be presented.

Response:

The oldest batch of CD19-CRE;UTX^{-/-} mice created during this project remained tumor free till 18 month. We included such information in the revised manuscript (Page 5). This batch of CD19-CRE;UTX fl/fl mice gave rise to the littermate populations used in tumorigenesis curve analysis (Fig. 1C-F). Since they were not part of the littermate, they were not plotted into the tumorigenesis curve.

2. Figure 1 E. In this figure, the authors compare male and female mice lacking UTX. However, there is already a considerable difference in tumor initiation between the E μ -Myc male and female mice. How does this influence the current data and the conclusions?

Response:

In order to assess the importance of UTX as an escape from X inactivation tumor suppressor, it is necessary to know whether UTY can prevent lymphomagenesis. The analysis in Figure 1 E, comparing the tumor development of UTX^{+/-} female and the UTX^{-/}Y male mice, provides an imperfect way to address this question. However, as pointed out by the reviewer, there are other factors in male and female mice that could influence lymphoma development. The most precise way to address whether UTY possesses tumor suppressor function is to make UTX;UTY double conditional KO mice. This is indeed a caveat that cannot be fully addressed by our current data. We added relevant discussion in the manuscript to clarify this point (Page 11-12).

3. The authors claim that loss of UTX results in advanced lymphoma, including brain metastasis, which caused skull bulges, and huge thymic/mediastinal tumors. However, and as also mentioned by the authors, i believe that the data rather shows that loss of *Utx* causes a switch in malignancy rather than metastasis of late-stage lymphoma. As characterised by the authors, the tumors might reflect a different

type of B-cell leukemia/lymphoma. I would rather say that loss of UTX results in more leukaemic features (instead of defining it as metastasis).

Response:

We think the malignancies we observed in our experiments are lymphomas based on the following reasons. First, in all E μ -Myc;UTX KO mice, including those mice that showed disseminations to brain and thymus, we observed significantly swollen bulges in nearly all lymph nodes. Human lymphomas specifically affect the lymph nodes, whereas leukemia do not typically form such solid tumors, especially in lymph node. Second, in most cases of human DLBCL, extranodal involvements are observed. Therefore we believe the malignancies in E μ -Myc;UTX KO mice are a more aggressive form of lymphomas, but not leukemia.

We realized that in the original manuscript we didn't specifically address the overall tumor presentation of those mice with brain/thymus disseminations, and it will cause questions about their disease identity. We added such information to the revised manuscript (Page 7) and we thank the reviewer for pointing this out.

4. The authors mention the occurrence of huge thymic tumors upon loss of UTX. Did the authors analyse the thymic tumor material to check if these tumor cells were from B cell origin?

Response:

The thymic tumors are from B cell origins. In all mice that we analyzed tumors with surface marker staining, lymphoma cells from major tumor sites including bone marrow, spleen, lymph node and the extranodal sites (skull and thymic, if present) showed B cell markers. We thank the reviewer for this question and added relevant information to the manuscript (Page 7).

5. the authors performed transcriptomics on the mouse tumors obtained in this study. How did they perform these experiments? Did they sort tumor cells? From what tissue?

Response:

For the transcriptomics analysis, we dissected lymphomas from major tumor bulks in mice (both nodal and extranodal), meshed the tumors and FACS-sorted lymphoma cells. The tumors are mostly composed of lymphoma cells with more than 90% purity. These lymphoma cells are single cells, therefore we do not need trypsin digestion to separate them. After lymphomas are crushed with glass slides, the lymphoma cells can pass through filter upon centrifugation, whereas most non-tumor cell structures are filtered out. The filtered cells are treated with red cell lysis buffers to further remove red blood cells before FACS sorting. Lymphoma cells are very small, round cells and can be identified through forward scatter and side scatter in FACS. After sorting, their identities were

confirmed by further cell surface marker staining.

A description of these experiments is provided in the method section of revised version (Page 15).

6. Besides transcriptomics, at least some mechanistic insights should be provided in relation to epigenetic deregulation upon loss of UTX. UTX CHIPseq and H3K27me3 ChIPseq should be performed on WT and UTX KO tumors, to correlate with the observed changes in gene expression.

Response:

As pointed out by reviewer #1, given that the B cell lymphomas in WT and UTX KO tumors are at different developmental stages, we are concerned that the developmental stage will have major impacts on the epigenetic markers of these tumors. This will render it very hard to interpret the CHIP results as to whether they are caused by UTX or developmental stages. Moreover, there is a previous report in UTX-deficient cancer cell lines that loss of UTX does not alter global levels of H3K27 methylation (Ezponda et al, 2017). In addition, for the genes that show different expression patterns in WT and UTX KO tumors, their expression pattern may be a direct result of UTX-mediated demethylation on such genes; however it can also result from secondary changes. For example, UTX may mediate demethylation on a transcriptional factor, which in turn impact expression of certain genes. Due to these factors, we did not perform such CHIP analysis.

7. The Efnb1 experiment is largely correlative. this setup is not testing whether the UTX phenotype is partly due to Efnb1. For this, one should do a rescue experiment

Response:

A similar question was raised by review #1.

In our experiment, we observed increased Efnb1 expression in UTX KO lymphomas. Since the UTX KO or WT lymphomas can not be grown in tissue culture, we used the E μ -Myc;Arf null cell line and showed that Efnb1 overexpression alone can mimic the enhanced aggressiveness on UTX KO tumors, such as brain metastasis.

We have tried many methods to generate a UTX KO E μ -Myc;Arf cells, but ultimately couldn't do it. The E μ -Myc;Arf cells are small, floating cells, and is very hard to transfect. It is also very poorly infected by lentivirus. We've tried many conditions, however we cannot introduce Cas9 into these cells by transfection or lentiviral transduction. These cells are receptive to retrovirus infection, but the size of the Cas9 gene exceeds the limit of retroviral system. Therefore, although Efnb1 is sufficient for promoting tumor aggressiveness, we cannot generate UTX KO E μ -Myc;Arf cells to test whether Efnb1 is necessary for UTX KO's aggressive phenotype. In light of this, we made relevant changes in the manuscript. (Page 2, 9,12)

Minor point

8. In the second part of the result section. The authors state "This strongly supported the theory proposed by Dunford et al, and showed that UTX is a key candidate for the 'escape from X-inactivation tumor-suppressor' (EXITS) genes". However, this concept was first introduced by an earlier paper of Van der Meulen et al. (Blood 2015; reference 11)

Response:

We thank the reviewer for the clarification and have made changes accordingly (Page 6).

Reviewer #3 (Remarks to the Author):

Li et al report that UTX has a role as an X-inactivation tumor suppressor and accelerates tumor growth in a myc-associated lymphoma model. They suggest that UTX deletion promotes malignant cell dissemination and is associated with increased angiogenesis, and that this is associated with increased expression of Efnb1. They also suggest that UTX deficiency renders the tumor cells more sensitive to cytarabine therapy.

Specific comments -

1. X-chromosome gain is seen with similar frequency to X-chromosome loss in B-cell lymphoma and when present is typically only seen in a small percentage of cells. To confirm the relevance of UTX loss in the pathogenesis of lymphoma particularly because deletion of UTX in B-cells alone did not induce lymphoma, the authors should show decreased expression of UTX in most malignant cells in various lymphomas.

Response:

According to the “two-hit” theory of cancer development, two major cancer-driving events are needed to allow for cancer formation. As a potential tumor suppressor, UTX loss alone did not cause lymphoma, however, it doesn’t argue against a tumor-suppressor role for UTX. UTX KO significantly accelerated lymphoma formation in the presence of oncogene Myc, and it is consistent with the “two-hit” cancer development model. Such lack of phenotype is also observed for other tumor suppressors. For example, VHL KO mice do not develop kidney cancer, and a recent study showed that crossing VHL KO with BAP1 KO mice produced kidney cancers (**Ref:** Wang, S. S., et al. (2014). "Bap1 is essential for kidney function and cooperates with Vhl in renal tumorigenesis." *Proc Natl Acad Sci U S A* 111(46): 16538-16543.).

Moreover, there are many routes to human malignancies, and loss of UTX expression is only one such possible route. Therefore, we believe it will be unlikely that we can observe a universal decreased expression of UTX in most malignant cells in various lymphomas. Instead, we argue that the important finding is, the portion of human lymphomas with low UTX expression is of significantly poor diagnosis and our mouse modeling data is consistent with such findings.

2. The authors need to address the clinical relevance of the tumor model. They use a myc-driven model of lymphoma (which is more similar to a Burkitt lymphoma than mature lymphomas), yet they then study the prognostic value in a dataset of diffuse large B-cell lymphoma rather than Burkitt lymphoma. Because the UTX^{-/-} E μ -Myc mice have a leukemic phenotype (and UTX loss has been associated with ALL), the authors should also discuss why they did not use a leukemia patient database for clinical correlations.

Response:

A similar question is asked by reviewer #2.

We think the malignancies we observed in our experiments are lymphomas based on the following reasons. First, in all E μ -Myc;UTX KO mice, including those mice that showed disseminations to brain and thymus, we observed significantly swollen bulges in nearly all lymph nodes. Human lymphomas specifically affect the lymph nodes, whereas leukemia do not typically form such solid tumors, especially in lymph node. Second, in most cases of human DLBCL, extranodal involvements are observed. Therefore we believe the malignancies in E μ -Myc;UTX KO mice are a more aggressive form of lymphomas, but not leukemia.

We realized that in the original manuscript we didn't specifically address the overall tumor presentation of those mice with brain/thymus disseminations, and it will cause questions about their disease identity. We added such information to the revised manuscript (Page 7) and we thank the reviewer for pointing this out.

Furthermore, about the E μ -Myc mouse model:

A chapter named 'Preclinical Modeling in Lymphoid Malignancies' written by Richa Dawar and Francisco J. Hernandez-Ilizaliturri in book 'Non-Hodgkin Lymphoma' 2013 has a detailed description of the E μ -myc mouse model as following:

"The E μ -myc transgenic mouse carries the c-myc oncogene coupled to the immunoglobulin heavy locus (IgH) enhancer E μ and results in the development of aggressive pre-B-cell or B-cell lymphomas accompanied by lymphoblastic leukemia but not Burkitt's lymphoma (BL). B-cell lymphoma development in the E μ -myc model occurs in 100% of animals, but the onset of the disease is rather variable. Mori and colleagues (**Ref**) described two distinct tumor phenotypes in the E μ -myc model: (1) the first type arises earlier in the life span of the mice and is composed mainly of immature B cells resembling BL; (2) the second type develops late in life (400 days after birth) and is composed of mature B cells simulating DLBCL." (**Ref**: Mori S, Rempel RE, Chang JT et al (2008) Utilization of pathway signatures to reveal distinct types of B lymphoma in the E μ -myc model and human diffuse large B-cell lymphoma. *Cancer Res* 68:8525–8534)

Based on the above, we address the clinical relevance of our tumor model as following.

Both E μ -Myc mouse model and UTX^{-/-} E μ -Myc mouse model, are B-NHL malignant lymphoma, sharing features with human Burkitt lymphoma and human DLBCL lymphoma. In humans, B-NHL malignancies are mostly originated from late stage B cells, whereas the original E μ -Myc mice develop mostly pre-and pro-B cell lymphomas. Knocking out UTX in this background enabled the generation of lymphomas of late stage B cells.

3. The evidence that UTX deficiency is associated with an increased propensity to disseminate or increased angiogenesis is not convincing. Increased disease in the brain or other organs, and increased "blood infusion" in organs, seems to be more extensive disease due to the more aggressive growth pattern that has been allowed

to progress rather than a functional change in the tumor cells. It is likely that control cells would result in a similar picture if allowed to progress for a few more days. The authors need to do additional studies to show that the cells migrate and adhere differently and studies to show increased new vessel formation in the presence of UTX-deficient cells compared to controls.

Response:

In our experiment, both for the E μ -Myc;UTX+/+ and the E μ -Myc;UTX-/- group, the mice were sacrificed at similar tumor burden and only the E μ -Myc;UTX-/- group showed extensive aggressive phenotypes. Moreover, the E μ -Myc mouse lymphoma model has been extensively used in many studies and it rarely showed extranodal dissemination at terminal disease stage. Therefore, it is unlikely that the more aggressive disease phenotypes in the UTX KO mice are caused by different length of disease progression. Moreover, RNA seq of lymphoma samples showed that genes involved in cell adhesions are differentially expressed in UTX KO tumors, which is consistent with the enhanced metastasis phenotype.

Lymphomas from E μ -Myc;UTX KO cells rapidly die when cultured *in vitro*, therefore we can't set up experiments with regard to cell migration and angiogenesis. We are more focusing on uncover the downstream target genes including Efnb1 and study their relevance in human patients instead of the low UTX level or UTX-deficiency.

4. The authors need to reconcile the finding that UTX-deficient cells are associated with increased sensitivity to cytarabine with their finding that UTX loss is associated with a poor prognosis. Increased sensitivity to treatment should result in more patients been salvaged with standard cytarabine-containing chemotherapy.

Response:

Cytarabine is usually used in leukemia treatment but is not the front-line treatment for lymphomas, which are usually treated with the R-CHOP combination therapy (Rituximab, cyclophosphamide, doxorubicin, vinblastine and prednisone). Current treatments of most lymphomas usually don't involve cytarabine. Therefore, it is not contradictory that UTX-deficient cells are associated with increased sensitivity to cytarabine, and that UTX loss is associated with a poor prognosis. Our finding is also consistent with recent clinical trial results showing that high-dose cytarabine may provide benefit for aggressive B cell lymphomas.

Minor comment -

5. Ezponda et al (Cell Rep 2017) recently published that UTX loss promotes a more aggressive phenotype in myeloma (another mature B-cell process) and sensitizes cells to EZH2 inhibition. The authors should reference this publication and discuss whether UTX deficient cells in their model are more sensitive to EZH2 inhibitors.

Response:

We thank the review for pointing this out and referenced the study on page 10.

The UTX KO cells can not be grown in tissue culture, therefore we used E μ -Myc;Arf cells to test the EZH2 inhibitor EPZ-6438, which is in clinical trials. However, in order to kill E μ -Myc;Arf cells, we need to use nearly 300 μ M of EZH2 inhibitor. Such a high concentration is inconsistent with targeted drug-induced cell death, and suggests that this drug is not active in treating E μ -Myc;Arf cells, and cell death at 300 μ M of this drug may be result of nonspecific killing. The EZH2 inhibitor GSK343 used in the Ezponda et al study is also inactive in E μ -Myc;Arf cells. Of note, the E μ -Myc;Arf cells are highly death-prone, and most drugs, such as doxorubicine, vinblastine, 6-TG and Methotrexate kills such cells at 10-20 nM concentration.

A possible reason for this drug to be inactive in E μ -Myc;Arf is that, in the multiple myeloma study by Ezponda et al, EZH2 inhibition caused reduction in endogenous Myc expression level, which contributed to cell death. This may reflect that the endogenous Myc promoter may be modulated by EZH2 activity. In contrast, in E μ -Myc;Arf cells the Myc gene is mis-expressed under the IgH enhancer and our results show they are insensitive to EZH2 inhibitor.

REVIEWERS' COMMENTS:

Reviewer #1 (Remarks to the Author):

The authors have addressed nearly all my concerns and have presented an interesting study. I have one remaining issue with the survival analysis in human DLBCL based on UTX and EFNB1 presented in Figure 4. The authors have still not shown a multivariable analysis to demonstrate that EFNB1 level is independently prognostic after taking UTX level into account. Their rebuttal letter states that the p-value of the risk group separation is "slightly better" when EFNB1 is added to the model with UTX level, therefore this is an improved biomarker. This is not correct statistics. They need to perform a Cox proportional hazards analysis, or similar multivariable analysis method, to ask if, when controlled for UTX level, EFNB1 level remains prognostic. Put another way, if they separate the DLBCL data into UTX high and low, then generate survival curves based on EFNB1 high and low in those subgroups, they should see a statistically significant separation of the curves to make this claim.

However, I don't think finding that EFNB1 is an independent prognostic marker is necessary for the impact of this paper. But, as presented currently, they cannot state that EFNB1 adds to UTX level in predicting survival in DLBCL.

Reviewer #2 (Remarks to the Author):

The authors nicely addressed most of my comments and concerns. For some comments, attempts were made to address the question, but these efforts were not successful. Therefore, some data are still lacking. Nevertheless, this reviewer appreciates the effort that has been made for all the additional experiments and I have no further comments or questions regarding the manuscript, which is in my opinion now acceptable for publication in Nat Comm.

Reviewer #3 (Remarks to the Author):

The authors have adequately addressed most of my concerns. However, they have not really addressed the fact that UTX deficient cells are sensitive to cytarabine and yet UTX loss is associated with a poor prognosis. Contrary to the author's comments, cytarabine is a standard agent in many lymphoma therapies (HyperCVAD is commonly used as initial therapy for aggressive lymphoma/Burkitt lymphoma and not RCHOP, DHAP is a standard salvage therapy, and BEAM is standard for transplant. All of these regimens include cytarabine). It seems that the use of regimens that include cytarabine as salvage treatment for lymphoma should then reverse the poor prognosis of UTX loss and UTX loss should be a favorable feature.

REPOSENSE TO REVIEWERS' COMMENTS:

Reviewer #1 (Remarks to the Author):

The authors have addressed nearly all my concerns and have presented an interesting study. I have one remaining issue with the survival analysis in human DLBCL based on UTX and EFNB1 presented in Figure 4. The authors have still not shown a multivariable analysis to demonstrate that EFNB1 level is independently prognostic after taking UTX level into account. Their rebuttal letter states that the p-value of the risk group separation is “slightly better” when EFNB1 is added to the model with UTX level, therefore this is an improved biomarker. This is not correct statistics. They need to perform a Cox proportional hazards analysis, or similar multivariable analysis method, to ask if, when controlled for UTX level, EFNB1 level remains prognostic. Put another way, if they separate the DLBCL data into UTX high and low, then generate survival curves based on EFNB1 high and low in those subgroups, they should see a statistically significant separation of the curves to make this claim.

However, I don't think finding that EFNB1 is an independent prognostic marker is necessary for the impact of this paper. But, as presented currently, they cannot state that EFNB1 adds to UTX level in predicting survival in DLBCL.

Response:

We thank the reviewer for the time and kind evaluation of our paper. We looked at the EFNB1 data, and we agree with the reviewer that EFNB1 do not perform as strongly as UTX as independent predictor of survival. Indeed the difference in p values does not directly argue for UTX-low and EFNB1-high as an improved biomarker. In light of this, we removed the relevant panels in Figure 4 and corresponding parts in figure legend and main text. We thank the reviewer for correcting such statistical analysis.

Reviewer #2 (Remarks to the Author):

The authors nicely addressed most of my comments and concerns. For some comments, attempts were made to address the question, but these efforts were not successful. Therefore, some data are still lacking. Nevertheless, this reviewer appreciates the effort that has been made for all the additional experiments and i have no further comments or questions regarding the manuscript, which is in my opinion now acceptable for publication in Nat Comm.

Response:

We greatly appreciate the time and critiques from all reviewers, as it helped us presented our results more clearly and improved the manuscript.

Reviewer #3 (Remarks to the Author):

The authors have adequately addressed most of my concerns. However, they have not really addressed the fact that UTX deficient cells are sensitive to cytarabine and yet UTX loss is associated with a poor prognosis. Contrary to the author's comments, cytarabine is a standard agent in many lymphoma therapies (HyperCVAD is commonly used as initial therapy for aggressive lymphoma/Burkitt lymphoma and not RCHOP, DHAP is a standard salvage therapy, and BEAM is standard for transplant. All of these regimens include cytarabine). It seems that the use of regimens that include cytarabine as salvage treatment for lymphoma should then reverse the poor prognosis of UTX loss and UTX loss should be a favorable feature.

Response:

We thank the reviewer for this point. Giving that regimens such as HperCVAD, BEAM are used in treating lymphomas, it does seem contradictory that UTX-low lymphoma are associated with low survival rate. Prompted by this question, we further examined the clinical information of the lymphoma database (GSE10846) analyzed in this paper. Out of the 414 patients, 181 were treated with CHOP-like regimen and 233 were treated with R-CHOP-like regimen.

181 Clinical info: Chemotherapy: CHOP-Like Regimen

233 Clinical info: Chemotherapy: R-CHOP-Like Regimen

Therefore, this group of patients was likely not extensively treated with cytarabine, and UTX-low patients' poor survival is therefore consistent with our finding. This highlights that currently there are portions of lymphoma patients that are not adequately treated with cytarabine, and as pointed out by the reviewer, inclusion of cytarabine in regimen may improve the treatment outcome of UTX-low lymphoma. We added the clinical information to the manuscript in the last paragraph of Discussion. Again we really appreciate this question, since such information helps clear off potential source of confusion.